# Fatty Acid Ratios Versus Conventional Risk Factors in Stroke: Insights into Severe Disability and Mortality Outcomes

**DOI:** 10.3390/nu17091518

**Published:** 2025-04-30

**Authors:** Sebastian Andone, Farczádi Lénárd, Silvia Imre, Mihai Dumitreasa, Rodica Bălașa

**Affiliations:** 1Department of Neurology, ‘George Emil Palade’ University of Medicine, Pharmacy, Science, and Technology of Târgu Mureș, 540136 Târgu Mureș, Romania; rodica.balasa@umfst.ro; 2Ist Neurology Clinic, Emergency Clinical County Hospital Târgu Mureș, 540136 Târgu Mureș, Romania; mihai.du96@gmail.com; 3Center for Advanced Medical and Pharmaceutical Research, ‘George Emil Palade’ University of Medicine, Pharmacy, Science, and Technology of Târgu Mureș, 540136 Târgu Mureș, Romania; lenard.farczadi@umfst.ro (F.L.); silvia.imre@umfst.ro (S.I.); 4Department of Analytical Chemistry and Drug Analysis, Faculty of Pharmacy, ‘George Emil Palade’ University of Medicine, Pharmacy, Science, and Technology of Târgu Mures, 540136 Târgu Mures, Romania; 5Doctoral School, ‘George Emil Palade’ University of Medicine, Pharmacy, Science, and Technology of Târgu Mureș, 540142 Târgu Mureș, Romania

**Keywords:** ischemic stroke, fatty acids, prevention, disability, mortality, stroke care, nutrition, dietary supplements

## Abstract

**Objective**: This study aims to investigate the role of fatty acid ratios, specifically DHA/ARA and EPA/ARA, in predicting severe disability and mortality in stroke patients and compare these ratios with conventional risk factors such as age, sex, hypertension, diabetes, and dyslipidemia. **Methods**: A prospective study was conducted involving 298 consecutive acute ischemic stroke patients (within 72 h of onset). Fatty acid ratios were measured from plasma, and all patients’ evolution was followed through hospitalization. Binary logistic regression analysis was used to identify predictors of severe disability at discharge (Rankin 4–6) and in-hospital mortality, including fatty acid ratios and conventional risk factors. **Results**: A higher DHA/ARA ratio was associated with a reduced chance of severe disability (OR = 0.81), while a higher EPA/ARA ratio was associated with an increased chance of severe disability (OR = 1.70). Age was a significant factor, with older age (median 70 years) associated with a lower survivability chance (OR = 0.93) and a higher likelihood of severe disability when surviving. Fatty acid ratios did not significantly affect mortality outcomes. For male patients, EPA/AA ratios showed a powerful association with severe disability (*p* = 0.045), while no significant effect of fatty acids was observed in females. **Conclusions**: Fatty acids were significant predictors of severe disability in patients with acute ischemic stroke, independent of conventional risk factors, but without having any effect on in-hospital mortality. Age remained the only significant conventional risk factor predictor of outcome. Integrating fatty acid ratios alongside conventional risk factors may improve predictions of severe post-stroke disability, potentially guiding more personalized interventions for stroke patients.

## 1. Introduction

Ischemic stroke is the neurological condition with the highest rate of hospitalization, representing the most common manifestation of cerebrovascular diseases [1].

The World Stroke Organization (WSO) estimated in 2022 that ischemic stroke ranks second globally in terms of mortality, with a 70% increase in incidence from 1990 to 2019. This rise is mainly driven by population aging and the increasing prevalence of vascular risk factors in many regions, particularly in low- and middle-income countries [2,3].

Despite this, mortality associated with ischemic stroke has shown a significant decline in recent decades across various regions, attributed to improvements in primary prevention and acute treatment. In the United States, age-adjusted mortality decreased by 32% between 1990 and 2017, reflecting progress in managing risk factors such as hypertension and dyslipidemia, along with the widespread adoption of antiplatelet and anticoagulant therapies [4].

Similarly, in the European Union, the age-standardized stroke mortality rate declined by an average of 4.2% annually between 1996 and 2015, with the most pronounced reductions observed in Southern and Western Europe [5]. These decreases are attributed to improvements in acute ischemic stroke management, including the admission of patients to specialized stroke units and the utilization of thrombolysis or mechanical reperfusion therapy [6].

Modifiable conventional risk factors, such as hypertension, diabetes mellitus, dyslipidemia, and smoking, alongside non-modifiable factors like age and sex, are well documented in the literature regarding their role in ischemic stroke [7,8].

Modifiable risk factors are primary targets in stroke prevention strategies, as their presence not only increases stroke occurrence but can also worsen post-stroke disability and mortality outcomes [9].

Furthermore, the accumulation of multiple risk factors is associated with increased incidence and recurrence of events, exacerbating long-term disability and elevating the risk of death [10].

In addition to these established factors, emerging evidence suggests that fatty acid profiles play a critical role in stroke outcomes; a high dietary omega-6/omega-3 PUFA ratio has been linked to an increased risk of ischemic stroke [11]. Low omega-3 levels are associated with worse post-stroke functional outcomes [12].

Fatty acids are pivotal to cellular functions, ranging from transmembrane regulation and energy storage to cellular signaling [13].

Polyunsaturated fatty acids (PUFAs), such as docosahexaenoic acid (DHA), eicosapentaenoic acid (EPA), and arachidonic acid (AA), are categorized into subtypes based on their structural differences, such as omega-3 (DHA and EPA) and omega-6 (AA).

These PUFAs are essential for human health, offering both neuroprotective [14] and anti-inflammatory effects [15]. DHA is incorporated into the phospholipids of neuronal membranes and supports synaptic activity [16], while EPA and AA serve as precursors for eicosanoids, which regulate inflammatory responses [15]. An imbalanced omega-3/omega-6 ratio can promote inflammatory and thrombotic processes, thereby increasing the risk of cardio-cerebrovascular events [17,18].

However, it remains unclear whether such an imbalance in omega-6/omega-3 PUFAs also influences stroke outcomes (such as post-stroke disability or survival), as this issue has been underexplored [19].

Despite extensive research on conventional risk factors, the prognostic significance of fatty acid imbalances in acute stroke outcomes remains poorly understood, representing a gap that this study addresses. Based on this, our study aimed to explore whether plasma fatty acid ratios (specifically DHA/AA and EPA/AA) are associated with severe stroke outcomes (severe disability or increased mortality). We also compared the prognostic effect of these ratios against conventional risk factors (age, sex, hypertension, diabetes, dyslipidemia, and smoking).

Our hypothesis is that an imbalance between omega-3 and omega-6 fatty acids would be linked to worse stroke disability.

## 2. Materials and Methods

### 2.1. Study Population

We performed a prospective observational study at our neurology clinic. A total of 321 consecutive patients with acute stroke were screened over the study period, of whom 23 were excluded based on the criteria below. The final sample comprised 298 patients with acute ischemic stroke.

**Inclusion criteria:**
Patients aged ≥ 18 years old;Acute ischemic stroke confirmed by clinical evaluation and imaging;Admission within 72 h from symptom onset.

**Exclusion criteria:**
Hemorrhagic strokes;Stroke mimics diagnosed after admission;Lack of consent for data collection and/or blood sampling.

**Study period**: We included patients over a six-month interval.

The study adhered to the principles of the Helsinki Declaration, and we received approval from the Ethics Committee of the Emergency Clinical County Hospital (no. 28763/13.11.2018). We also obtained written informed consent from all patients, legal guardians, or family members to collect data and blood samples.

### 2.2. Data Collection

Clinical, paraclinical, and demographic data were collected from the patients’ charts and introduced into a specifically designed research database, which was used to process all the data once the collection phase had been completed.

Severe disability was defined as a modified Rankin Scale of 4–6 at discharge (scores of 0–3 were considered non-severe), with 6 indicating death.

### 2.3. Fatty Acid Analysis Process

After we received consent, we collected the blood samples within 24 h of admission. We used EDTA vacutainers with a gel separator (PRIMA Lab SA, Balerna, Switzerland), and after the samples had clotted at room temperature, we centrifuged them for 15 min using a rotational speed of 3500 rpm. After the centrifugation, the plasma was separated into cryotubes for storage at −40 °C.

To analyze the fatty acids AA, EPA, and DHA, we used a complex method involving high-performance liquid chromatography (HPLC) combined with mass spectrometry (MS). We used arachidonic-d11 acid as an internal standard reference. The samples were analyzed using reversed-phase liquid chromatography (LC), after which a mass spectrometer with an MRM MS/MS module was used for detection [20].

Standard solution preparation: We prepared fresh calibration standards using 0.2% formic acid, with a concentration of between 2.5 and 250 µg/mL for AA and of 50–2500 ng/mL for DHA and EPA. The samples were vortexed and centrifuged before they were aliquoted into HPLC vials.

Plasma sample processing: Fresh plasma samples were prepared using 200 µL of plasma, 100 µL internal standard, and 500 µL acetonitrile. Before samples were introduced into HPLC vials, the same process as in standard solution preparation was performed.

HPLC conditions: For the analysis, we used a Perkin-Elmer Flexar 10 UHPLC system (PerkinElmer, Inc., Shelton, CT, USA) with an XB-C18 Kinetex column (Phenomenex, Inc., Torrance, CA, USA). For the mobile phase, we used a solution of 15% ammonium formate and 85% acetonitrile, which was introduced at a 0.4 mL/min rate. During the analysis, the column temperature was 25 degrees, and the whole process took around 6 min.

Mass spectrometry: We used a Sciex 4600 QTOF mass spectrometer (SCIEX, Framingham, MA, USA) following negative-ion spray ionization to detect AA, EPA, DHA, and the internal standard. We used specific ion transitions.

We used a linear calibration model with 6 levels for AA and 5 levels for DHA and EPA, each weighing 1/y2.

### 2.4. Graphics

We created all figures using online tools such as https://www.codabrainy.com/en/python-compiler/ (accessed on 9 September 2024), Python-language programming (Python 3.6), and Adobe Photoshop CS4.

### 2.5. Statistical Analysis

Continuous variables are presented as mean ± standard deviation (SD) and were compared between groups using Student’s *t*-tests (two-tailed), using a *p*-value of ≤0.05 as statistically significant. Categorical variables were compared using the chi-square test.

We created two binary logistic regression models to identify independent predictor variables of severe disability at discharge and in-hospital mortality. Severe disability was defined as mRS 4–6 at discharge and in-hospital death as 6 at discharge. The model’s performance was evaluated using IBM SPSS 26, with a cut-off value of 0.5. Model fit was assessed using the Hosmer–Lemeshow test. The following predictors were entered into the logistic regression model: age, hypertension, dyslipidemia, diabetes, smoking status, DHA/AA ratio, and EPA/AA ratio.

Statistical analysis was performed using IBM SPSS Statistics v26 and Microsoft Excel 2019.

## 3. Results

### 3.1. Demographic, Risk Factors, Disability, and Mortality—Population Analysis

We included a total of 298 patients, with a male-to-female ratio of 1.05:1. The average age was 67.27 ± 14.14 years in male patients versus 72.74 ± 12.11 years in female patients (*p* < 0.001). Despite this age difference, the mean modified Rankin Scale (mRS) did not differ significantly, with 2.30 ± 1.73 in males vs. 2.70 ± 1.70 in females (*p* = 0.061).

Regarding mortality, the proportion of patients who died during hospitalization was similar between genders, with 9 males and 10 females, resulting in no significant difference (*p* = 0.814).

The analysis of fatty acid ratios revealed no significant gender differences in the levels of DHA/AA (*p* = 0.179) or EPA/AA (*p* = 0.773), suggesting that these fatty acid ratios were relatively consistent between male and female patients.

When examining conventional risk factors, significant gender differences were observed in hypertension, dyslipidemia, and smoking status. Although the number of hypertensive male and female patients was mostly similar, the number of patients without hypertension was significantly higher in the male group (*p* = 0.029).

As for dyslipidemia, the prevalence was higher in the female group (*p* = 0.047).

Smoking was much more common in males, with 57 males reporting a history of smoking compared to only 16 females (*p* < 0.001).

On the other hand, diabetes was not significantly different between genders (*p* = 0.240), as both groups had a similar proportion of patients with this condition.

All the data mentioned above are summarized in Table 1.

Comparing the two groups, we can see that a higher proportion of males (113) were in the low-to-moderate disability group (Rankin scores 0–3) compared to females (88), while more females had severe disability (Rankin scores 4–6) compared to males, with 40 males and 57 females categorized as severely disabled. This difference was statistically significant (*p* = 0.007) (Figure 1).

### 3.2. Stroke Severe Disability Prediction Models

The logistic regression analysis for all stroke patients revealed several significant associations with severe disability.

From all conventional risk factors, only age was significantly associated with an increased likelihood of severe disability (B = −0.044, *p* < 0.001). Each additional year of age increased the odds of severe disability by approximately 4% (OR = 0.957).

The DHA/AA ratio was associated with lower odds of severe disability (OR = 0.809, 95% CI 0.68–0.97, *p* = 0.028), while the EPA/AA ratio was associated with higher odds of severe disability (OR = 1.698, 95% CI 1.07–2.70, *p* = 0.026).

Other variables did not show significant associations and were not found to influence the odds of severe disability significantly (Figure 2).

The logistic regression analysis for female stroke patients revealed that age was significantly associated with an increased likelihood of severe disability (B = −0.041, *p* = 0.016, OR = 0.960). Other traditional risk factors, including hypertension and smoking, did not significantly influence the likelihood of severe disability in this cohort of female patients (Figure 3).

In the male subgroup, the EPA/AA ratio remained a significant predictor (OR = 2.306, *p* = 0.045, B = 0.835), while in the female subgroup, both fatty acid ratios were insignificant (*p* > 0.10 for both) (Figure 3 and Figure 4).

Age was significantly associated with an increased likelihood of severe disability (B = −0.050, *p* = 0.009, OR = 0.951).

The EPA/AA ratio was positively associated with severe disability (B = 0.835, *p* = 0.045), increasing proportionally the likelihood of severe disability (OR = 2.306).

Although the DHA/AA ratio showed a negative association with severe disability, it was not statistically significant (*p* = 0.094).

Other risk factors did not significantly influence the likelihood of severe disability in the male cohort (Figure 4).

### 3.3. Stroke Mortality Prediction Models

In the logistic regression analysis for mortality regarding all stroke patients, age was found to increase the likelihood of death significantly, with each additional year decreasing the odds of survival by approximately 7% (B = −0.069, *p* = 0.011, OR = 0.934). This suggests that older stroke patients were less likely to survive compared to younger patients.

However, none of the other variables, including fatty acid ratios (DHA/AA and EPA/AA) or traditional risk factors such as diabetes, hypertension, dyslipidemia, and smoking, were significantly associated with mortality (Figure 5).

## 4. Discussion

### 4.1. Sex-Specific Differences

Our subgroup analysis showed sex-specific differences, even if the study population had similar sex distribution (153 male vs. 145 female patients). In male patients, the EPA/AA ratio showed a greater association with disability risk (DHA/AA had a similar trend), while in female patients, there was no significant association between fatty acid ratios and disability. The studies in the literature support these findings. Women experience worse stroke outcomes than men more frequently, due to risk factors such as older age at stroke onset and having more stroke comorbidities [21,22]. Some authors even suggest that stroke severity patterns are different in women (e.g., lesion distribution) [23]. Notably, our female cohort had conventional factors (including age) as primary determinants of outcome, potentially reducing any effect of fatty acids. In contrast, the male patients (who were younger) showed greater influence of fatty acids, as the EPA/AA had a clear impact on disability.

Even if the mean value of modified Rankin scores between the two sex subgroups was not significant, when we compared the different classes regarding disability, we saw that female patients were more likely to develop severe disability than men. When comparing the results of the prediction models for all patients to the male and female cohorts, age remained the only significant predictor of severe disability across all groups, with a slightly more substantial effect in males. As some authors suggested, this finding may be because women experience stroke at an older age than men, which may impact increased disability due to age-related health factors [22]. Other studies have reached the same conclusion, pointing out that other factors, except older age, such as greater stroke severity and pre-stroke health issues, contribute to worse outcomes [24,25].

Not only the short-term prognosis but also the long-term prognosis seems to be different in female patients, as shown from a meta-analysis that concluded that female patients have higher 1-year and 10-year mortality and higher recurrence rates compared to male patients [26]. Another mechanism that may contribute to women having greater disability and worse outcomes could be explained by factors such as hormonal and genetic variations [27]. The American Heart Association’s statistics also show that the risk of stroke increases exponentially with each decade of life, also in addition to correlating with worse outcomes [28]. Another study also showed that increased age plays a significant role in increased rates of stroke severity and mortality [29]. All these findings support our results, as age was a clear determining factor for increased chances of poor outcomes, with severe disability and higher mortality being non-dependent on the sex of the patient.

When comparing the logistic regression results between male and female stroke patients, several key differences emerge regarding the predictors of severe disability. Regarding the fatty acid ratios, the results show gender-specific differences. In females, both the DHA/AA ratio and the EPA/AA ratio had a significant impact on severe disability. While the DHA/AA ratio was negatively associated with severe disability, the EPA/AA ratio was positively associated with severe disability. However, in males, only the EPA/AA ratio had a significant positive association with severe disability, although with a much stronger effect than in the female group.

### 4.2. Fatty Acids—Biological Mechanisms

Our findings raise the possibility that EPA and DHA could have divergent influences on stroke outcome: A higher EPA/AA ratio was associated with more severe disability, whereas a higher DHA/AA ratio appeared protective. This is biologically plausible—EPA may promote a pro-inflammatory milieu under certain conditions, whereas DHA exerts anti-inflammatory and neuroprotective effects [14,15].

One study emphasizes that specific fatty acid balances, such as a higher DHA/AA ratio, may influence stroke severity. This finding strengthens our study’s findings, which show that this ratio increased the likelihood of a more severe disability at discharge [30]. Supporting our findings on how fatty acids did not influence the disability outcome in the female group, findings in the literature show that women have higher baseline levels of DHA caused by fatty acid metabolism and storage, and sex-specific differences [31,32].

This might mean that female patients have better compensation mechanisms regarding fatty acid fluctuations, and the outcomes might be less noticeable in females. Another theory that could explain why the fatty acid ratios do not influence women’s disability might be related to how the immune system responds differently in women compared to men. This might also extend to how fatty acids influence and modulate inflammation, as in women, this mechanism could be much more diminished than in men [33,34].

Notably, although both EPA and DHA are omega-3 PUFAs, our results and other studies suggest they should not be viewed interchangeably. DHA appears to have a more direct neuroprotective role in the brain, whereas EPA’s impact may be mediated through inflammation. Thus, studies that report overall “omega-3” effects on stroke need to be interpreted in light of the distinct roles of DHA vs. EPA [15,35].

### 4.3. Omega-3 PUFAs and Stroke: Clinical Evidence

An investigation of the relationship between PUFA ratios and neurological deterioration in acute ischemic stroke found that a low omega-3/omega-6 ratio is associated with greater neurological decline. The authors also suggested that this ratio may be used as a predictive biomarker for stroke progression [36].

A review on the roles of omega-3 PUFAs showed that they can help mitigate post-stroke-related injury. They showed that omega-3 fatty acids can rapidly modulate inflammation, supporting the idea that they have an active role in the dynamic changes of inflammation and repair. This suggests that factors related to metabolism and inflammation, which can change dynamically, might be more important for recovery than conventional risk factors. The direct action in neuronal repair and inflammation supports the fact that fatty acids act as rapid response modulators and have an impact on disability severity [37].

Another study indicated that DHA has an essential role in neuronal survival and in the capacity to recover from injury [38]. Additionally, the roles of omega-3 PUFAs in neuroprotection and neurotransmission were highlighted [39]. In experimental models, the administration of DHA reduced the infarct volume and promoted neurogenesis and angiogenesis after stroke [35]. These authors’ findings could explain why, compared to the conventional risk factors that have a role in the initial vascular event, fatty acids might also have a role in the aftermath of the stroke, having a dual role in both prevention before the event and neuroprotection after the event.

A meta-analysis of over 180,000 patients showed that increased serum levels of omega-3 PUFAs are associated with decreased risk of total and ischemic strokes. Still, there was no change regarding the incidence of hemorrhagic strokes. This might suggest that the protective effect is focused only on specific stroke types [40].

The therapeutic effects of omega-3 PUFAs following stroke events were studied using animal models; they demonstrated that post-stroke administration of PUFAs improved the neurological outcomes [41]. The same authors also showed in another study that this therapeutic procedure enhanced angiogenesis and neuron survival. They pointed out, though, that this treatment’s efficacy decreased proportionally with age, suggesting a stronger benefit in younger patients [42].

In a study examining the relationship between omega-3 serum levels and stroke severity at admission and 3-month prognosis, the authors concluded that higher serum levels of omega-3 PUFAs are associated with milder stroke severity and better functional outcomes [12]. On the other hand, lower ratios of EPA/AA and DHA/AA could contribute to the progression of ischemic stroke, especially in young patients. This suggests that maintaining balanced omega-3 and omega-6 PUFA ratios can help prevent stroke risk and reduce the severity of stroke [43].

### 4.4. Fatty Acids vs. Conventional Risk Factors

For other risk factors, including diabetes, hypertension, dyslipidemia, and smoking, no significant associations were found in either the sex subgroup or in the combined study group. When comparing the ratio of fatty acids with conventional risk factors, we need to point out that age is universally significant, being a marker that reflects cerebrovascular changes.

However, conventional risk factors such as hypertension, diabetes, dyslipidemia, and smoking, despite increasing the incidence of stroke, do not reflect the status of the acute inflammatory response and the brain injury severity after the occurrence of a stroke event [44]. A pooled analysis showed that patients with higher serum levels of omega-3 had a lower incidence of ischemic stroke, even after adjusting for conventional risk factors. This was independent of blood pressure or lipid levels [40].

In our previous study, we also showed similar findings regarding conventional risk factors and fatty acid profiles. Both smoking and dyslipidemia were found to influence serum levels of fatty acids or their ratios, potentially increasing stroke risk through fatty acid-mediated mechanisms [45].

The direct impact of omega-3 fatty acids on pro-inflammatory pathways can reflect real-time tissue injury much better than common risk factors like hypertension or diabetes. This is because stroke damage is directly linked to acute inflammation, in which fatty acid balance plays a role, while other conventional risk factors do not [15].

It has also been emphasized that although conventional risk factors like dyslipidemia and hypertension are determinants in stroke incidence and subtype, the role of inflammatory and neuroprotective factors might predict the severity of disability better than those previously mentioned [44].

All these arguments provide a strong rationale for why fatty acids like DHA and EPA, with their well-documented roles in inflammation and neuronal activity processes, might be key determinants of disability, given their close relationship with the acute pathophysiology involved in stroke injury.

### 4.5. Limitations

This study was conducted in a single center with a moderate sample size (n = 298), which may limit its general applicability. Our outcome assessment was based on short-term follow-up (disability at discharge); we did not track long-term disability, so we cannot state whether fatty acid levels predict long-term outcomes in any way.

Additionally, regardless of whether we found associations, the observational design cannot prove causation, as factors that have not been measured, such as treatment or diet prior to stroke, might have influenced our results.

Finally, our measurements are based on fatty acid levels on admission, and we did not follow up on further changes in plasma levels either during hospitalization or after.

## 5. Conclusions

In summary, our study shows that fatty acid ratios (DHA/AA and EPA/AA) are independent predictors of stroke disability, although they do not influence in-hospital mortality.

Conventional risk factors, primarily age, remain essential in outcome prediction; however, other factors, like hypertension, diabetes, and dyslipidemia, were not associated with severe disability in our cohort.

These findings suggest that evaluating a patient’s omega-6/omega-3 balance upon admission could increase prognostic precision for functional outcomes.

Further research is needed to validate these relationships and to explore the main mechanisms, but our results suggest a potential role for metabolic biomarkers, like fatty acids, in stroke outcomes.

## Figures and Tables

**Figure 1 nutrients-17-01518-f001:**
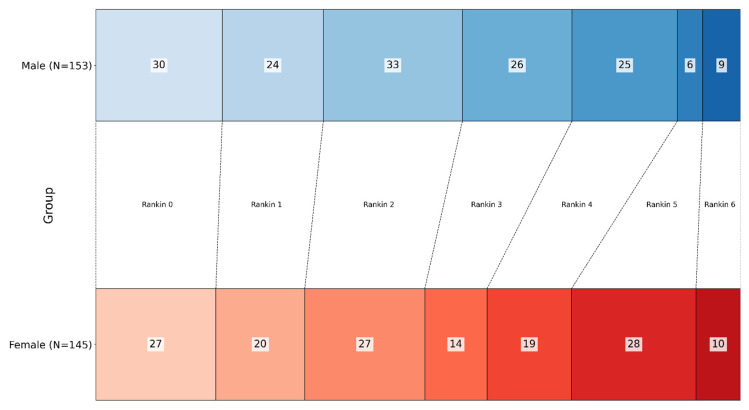
Distribution of modified Rankin Scale (mRS) scores at discharge, grouped by sex. Male patients had a higher proportion of good outcomes (mRS 0–3), while female patients had a higher proportion of severe disability (mRS 4–6). This difference was statistically significant (*p* = 0.007). (Gradient blue represents male patients, gradient red represents female patients; each shade corresponds to the respective Rankin score).

**Figure 2 nutrients-17-01518-f002:**
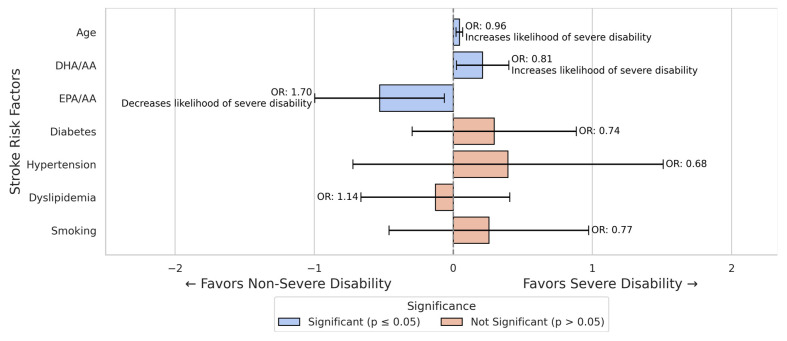
Logistic regression results for predictors of severe disability in all stroke patients. The model identified older age and DHA/AA ratio as factors associated with increased odds of severe disability. In contrast, a higher EPA/AA ratio was associated with increased odds of severe disability.

**Figure 3 nutrients-17-01518-f003:**
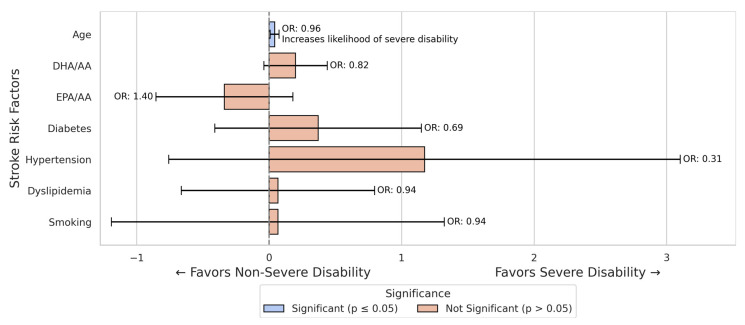
Logistic regression results for predictors of severe disability in female stroke patients. In this model, age was the only significant predictor of severe disability, while neither DHA/AA nor EPA/AA ratio showed a significant effect on disability outcomes.

**Figure 4 nutrients-17-01518-f004:**
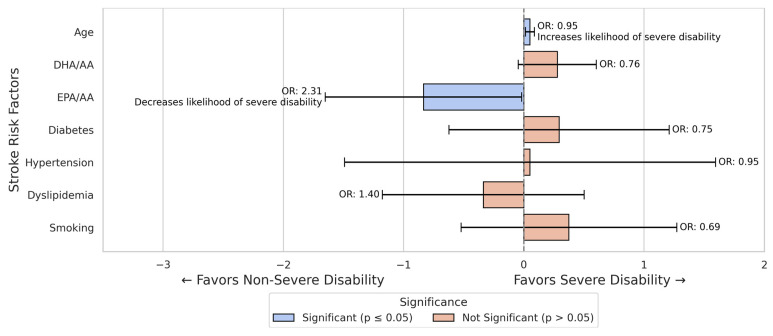
Logistic regression results for predictors of severe disability in male stroke patients. In this model, age was a significant predictor of severe disability, while the EPA/AA ratio was associated with lower odds of severe disability. Notably, DHA/AA was not significantly associated with the outcome.

**Figure 5 nutrients-17-01518-f005:**
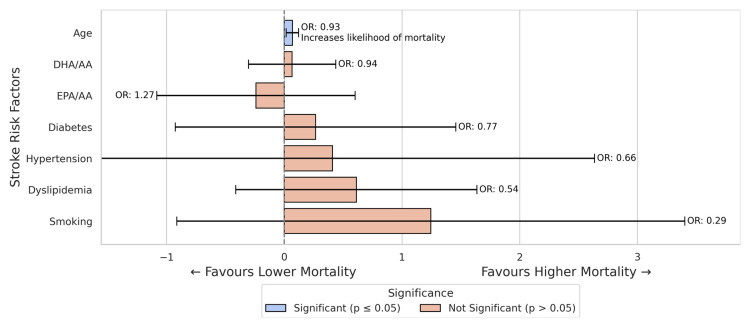
Logistic regression results for in-hospital mortality in all stroke patients. In this model, age was a significant predictor of mortality (older patients had lower chances of survival). At the same time, fatty acid ratios (DHA/AA, EPA/AA) and other conventional risk factors were not significantly associated with in-hospital death.

**Table 1 nutrients-17-01518-t001:** Clinical and demographic characteristics.

*n* = 298	Male (*n* = 153)	Female (*n* = 145)	
Age	67.27 ± 14.14 (25–100)	72.74 ± 12.11 (32–94)	*p* < 0.001
Rankin at discharge	2.30 ± 1.73	2.70 ± 1.70	*p* = 0.061
0	30	27	
1	24	20	
2	33	27	
3	26	14	
4	25	19	
5	6	28	
6 (Dead)	9	10	
**Disability**			
Low–moderate (Rankin 0–3)	113	88	*p* = 0.007
Severe (Rankin 4–6)	40	57	
**Mortality**			
Alive at discharge	144	135	*p* = 0.814
Died during hospitalization	9	10	
Fatty acid ratios			
DHA/AA	4.96 ± 2.31	5.04 ± 2.13	*p* = 0.179
EPA/AA	1.79 ± 1.13	1.75 ± 1.19	*p* = 0.773
**Conventional risk factors**	**Yes/No**	**Yes/No**	
Hypertension	136/17	139/6	*p* = 0.029
Diabetes	36/117	43/102	*p* = 0.240
Dyslipidemia	77/76	90/55	*p* = 0.047
Smoking	57/96	16/129	*p* < 0.001

**Abbreviations:** DHA = docosahexaenoic acid; EPA = eicosapentaenoic acid; AA = arachidonic acid. Age is presented as mean ± standard deviation, with values in parentheses indicating the range (min–max).

## Data Availability

The data presented in this study are available on request from the corresponding author due to privacy.

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
