# Peer review of "Fatty Acid Ratios Versus Conventional Risk Factors in Stroke: Insights into Severe Disability and Mortality Outcomes"

_nutrients, 2025, doi:10.3390/nu17091518_

Round 1
Reviewer 1 Report
Comments and Suggestions for Authors
INTRO
- Pls add a sentence to explain where and why stroke incidence is raising beginning Intro
- Unclear “Modifiable risk factors are primary targets in preventive strategies for ischemic stroke, as they can directly 61 impact disability and mortality rates”. Please add reference about stroke disability since the literature is mainly about risk factors and stroke risk.
- “In addition to these established risk factors, less-explored contributors such as fatty 65 acids also play a critical role”. Where are the references about this statement?
- “These 70 PUFAs are essential for human health, offering both neuroprotective and anti-inflamma- 71 tory effects”. Where are the references about this statement?
- “DHA is incorporated into the phospholipids of neuronal membranes and sup- 72 ports synaptic activity, while EPA and AA serve as precursors for eicosanoids, which reg- 73 ulate inflammatory responses”. Where are the references about this statement?
- “An imbalanced omega-6 to omega-3 ratio can promote inflammatory and thrombotic 75 processes, thereby increasing the risk of cardio-cerebrovascular events. [11, 12]”.
- Do you mean Omega 3/Omega 6 ratio?
- This statement “An imbalanced omega-6 to omega-3 ratio can promote inflammatory and thrombotic 75 processes, thereby increasing the risk of cardio-cerebrovascular events” does not define the gaps of knowledge behind the aim of the study that is about stroke outcome and not stroke risk. Provide refs (if available) about imbalanced omega3/6 ratio in relation with stroke outcome and connect the Intro with the study aim
- Explain what is the rational behind the fact that imbalance of omega6/Omega 3 should be bad for outcome. What the literature reports about this? What are the mechanisms to explain this?
- Develop an Aim end of Intro
M&M
- Pls check whether possible to have M&M section after Discussion according to the rules of the journal
DISCUSSION
The Discussion is hard to follow
- “These findings may hint that elevated EPA may contribute to an unfavorable out- 205 come related to a pro-inflammatory mechanism, while DHA might have the opposite ef- 206 fect, given its potential neuroprotective and anti-inflammatory properties”. Where are the references about this statement?
- The data suggest a positive role of DHA and a negative role of EPA (both being Omega 3). However, the discussion report many reference about “omega 3” effects on stroke outcome (and risk) without discussing what this means considering that in Omega 3 there are both EPA and DHA…The text should be revised to be more stringently discussed in relation to the obtained results.
- The whole discussion should be cleaned up from reporting effects of FFA etc on stroke risk. This work is about outcome and the discussion should focused on outcome
- The whole discussion is unclear and hard to follow. The authors should scrutinize first the results about the 2 ratio and explain them in details based on the literature (what is the meaning that the 2 ratio are associated with different outcome?). Then males/versus females should be discussed. The whole should not include literature about stroke risk that is irrelevant in relation to stroke outcome/prognosis
Author Response
- Comment: “Please add a sentence to explain where and why stroke incidence is rising (beginning of Introduction).”
Response: We have added a sentence explaining where (globally, especially in low- and middle-income countries) and why stroke incidence is rising (due to demographic changes and higher burden of risk factors). This addresses the reviewer’s concern by providing context at the beginning of the Introduction. - Comment: “Unclear: ‘Modifiable risk factors are primary targets in preventive strategies for ischemic stroke, as they can directly impact disability and mortality rates.’ Please add a reference about stroke disability since the literature is mainly about risk factors and stroke risk.”
Response: We agree that this statement required support. We have revised it to explicitly link modifiable factors with stroke outcomes and offered the reference to support it. - Comment: “‘In addition to these established risk factors, less-explored contributors such as fatty acids also play a critical role.’ Where are the references about this statement?”
Response: We have strengthened this statement by citing key studies. These additions provide evidence that fatty acids are indeed relevant to stroke, thus justifying our focus on fatty acid ratios. - Comment: “‘These PUFAs are essential for human health, offering both neuroprotective and anti-inflammatory effects.’ Where are the references about this statement?”
Response: We have provided supporting literature for this statement. These references support our claim that PUFAs have neuroprotective and anti-inflammatory effects. - Comment: “‘DHA is incorporated into the phospholipids of neuronal membranes and supports synaptic activity, while EPA and AA serve as precursors for eicosanoids, which regulate inflammatory responses.’ Where are the references about this statement?”
Response: We added two references to confirm these mechanistic facts. Including these citations makes the biochemical roles of DHA, EPA, and AA clear and well-supported. - Comment: “‘An imbalanced omega-6 to omega-3 ratio can promote inflammatory and thrombotic processes, thereby increasing the risk of cardio-cerebrovascular events. [11, 12]’. Do you mean Omega-3/Omega-6 ratio?”
Response: Yes, the ratio is indeed Omega-3/Omega-6. To address the reviewer’s question, we explicitly clarified this issue in the text. - Comment: “This statement ‘…increasing the risk of cardio-cerebrovascular events’ does not define the gaps of knowledge behind the aim of the study, which is about stroke outcome and not stroke risk. Provide refs (if available) about imbalanced omega-3/6 ratio in relation to stroke outcome and connect the Intro with the study aim.”
Response: We acknowledge the reviewer’s point that the knowledge gap needed to be made explicit. We have now explicitly stated that the impact of omega-6/omega-3 imbalance on stroke outcomes is not well established. This change clearly identifies the gap that our study addresses and sets the stage for our research objective. - Comment: “Develop an Aim at the end of Introduction.”
Response: We have now clearly stated the objective and scope of our study at the end of the Introduction. This new paragraph directly addresses the identified gap by outlining that we will examine fatty acid ratios in relation to stroke outcomes and compare them to traditional risk factors. We also explicitly mention our hypothesis that an unfavorable omega-6/omega-3 balance would correspond to poorer outcomes. This addition fulfills the request for a clear research aim and provides the reader with our study’s purpose upfront. - Comment: “Materials & Methods: Please check whether it’s possible to have the M&M section after Discussion according to the rules of the journal.”
Response: We reviewed the author instructions for the journal and moved the M&M accordingly. This re-ordering aligns the manuscript with the journal’s format and the reviewer’s suggestion. - Comment: “Discussion: The Discussion is hard to follow.”
Response: We understood the reviewer’s general concern and undertook a substantial rewrite of the Discussion section. The revised Discussion is more structured: it first summarizes the key results, then discusses them in the context of existing literature (focusing on outcome studies rather than risk incidence) and addresses the sex-specific observations separately. We believe these changes make the Discussion easier to follow. - Comment: “‘These findings may hint that elevated EPA may contribute to an unfavorable outcome… while DHA might have the opposite effect…’ – Where are the references about this statement?”
Response: We addressed this by providing literature-based reasoning and references. We also explicitly note this as a hypothesis consistent with known biology, rather than a definitive conclusion, to avoid unsupported claims. - Comment: “The data suggest a positive role of DHA and a negative role of EPA (both being Omega-3). However, the discussion reports many references about ‘omega-3’ effects on stroke outcome (and risk) without discussing what this means considering that in Omega-3 there are both EPA and DHA… The text should be revised to be more stringently discussed in relation to the obtained results.”
Response: In the revised text, we explicitly discuss DHA and EPA separately in the context of our findings. By doing so, the discussion directly addresses the apparent paradox of “omega-3 benefits” in general literature versus the divergent roles we observed. - Comment: “The whole discussion should be cleaned up of reporting effects of FFA (free fatty acids) etc. on stroke risk. This work is about outcome and the discussion should focus on outcome.”
Response: The revised discussion avoids drifting into stroke incidence or prevention literature except where absolutely relevant. We agree that including broad stroke risk factors and prevention studies was distracting. The discussion now predominantly cites studies on stroke recovery, severity, and prognosis. - Comment: “The authors should scrutinize first the results about the 2 ratios and explain them in detail based on the literature (what is the meaning that the 2 ratios are associated with different outcome?). Then male versus female differences should be discussed. The whole should not include literature about stroke risk that is irrelevant to outcome/prognosis.”
Response: We have followed the reviewer’s guidance on structure. The Discussion now first addresses the two fatty acid ratios and their implications, supported by literature (e.g., mechanistic roles of DHA vs. EPA as discussed). After thoroughly examining those results, we then address the sex differences observed. We combined the previously fragmented sentences about sex-specific findings into a unified discussion. We believe this restructuring makes the discussion clearer and more directly relevant to our results.
Reviewer 2 Report
Comments and Suggestions for Authors
The manuscript has a relevant topic but several major issues need to be addressed. The manuscript is not ready for publication and it should be written completely in introduction, results, and discussion. In addition, the methods section should be improved too.
I provide detailed comments below to help the authors improve their work.
- The abstract is weak and needs to be strengthened.
- The abstract lacks specificity regarding the methodology (sample size, inclusion/exclusion criteria).
- The abstract results section is weak.
- No concrete numerical results are provided, making it difficult to gauge the study’s findings.
- The conclusion in the abstract is overly broad and not directly related to specific evidence. What is the take home message? I don’t see any specific message.
- The introduction is too general and lacks a clear research gap analysis. What is the contribution of this paper? Where is the misunderstanding in the field? How this can add value?
- What is the hypothesis of this paper? What is the aims and objectives of this manuscript?
- The manuscript needs some language and grammar editing.
- The manuscript has too many paragraphs. For example, in Discussion section, some paragraphs are only 2, 3 lines (lines 280-310). This breaks the flow of the manuscript.
- Table 1: age range should be provided.
- Figure legends are too short and not informative.
- The methods section is poorly written. The methodology lacks clarity and reproducibility.
- The inclusion and exclusion criteria are not detailed enough.
- The statistics section is not clear.
- The results are presented in a vague manner without sufficient numerical detail.
- The discussion is largely speculative and lacks a connection to the results.
- The discussion should discuss the obtained results and make connections with available literature by comparison and contrast. No comparison with previous literature or explanation of how findings align or differ.
- The manuscript needs a limitations paragraph to explain the limitations.
- The conclusion is too broad and doesn’t really connect to the results. It makes claims that are not fully supported by the data.
- Some landmark and important papers in the field are not cited.
Author Response
Reviewer 2
- Comment: “The abstract is weak and needs to be strengthened.”
Response: We acknowledged that the original abstract lacked impact. The revised abstract now contains the essential details and findings. The methodology is described with greater specificity, the results section includes key quantitative findings, and the conclusions clearly state the main message. These improvements address the weaknesses identified by the reviewer. - Comment: “The abstract lacks specificity regarding the methodology (sample size, inclusion/exclusion criteria).”
Response: We have now explicitly stated the sample size (n=298), the population (acute ischemic stroke patients, within 72h of onset), and the outcomes analyzed. We also implicitly indicate inclusion criteria (acute ischemic stroke, timeframe) in the abstract. By mentioning the outcome measures and analysis method, the abstract’s methodology section is far more specific. - Comment: “The abstract results section is weak. No concrete numerical results are provided, making it difficult to gauge the study’s findings.”
Response: We introduced key statistics to make the results clear and quantitative. This includes odds ratios (ORs) with confidence intervals for the effect of the fatty acid ratios on severe disability, as well as a note on the p-value for age and for sex-specific findings. - Comment: “The conclusion in the abstract is overly broad and not directly related to specific evidence. What is the take-home message? I don’t see any specific message.”
Response: This revised conclusion is much more specific. It explicitly states what we found (that fatty acid ratios predict disability but not mortality, and that age is still very important) and the implication (that assessing fatty acid ratios could enhance outcome prediction). The take-home message is now clear: fatty acid ratios matter for stroke prognosis (disability), beyond the information provided by classic risk factors. This is directly supported by our results. - Comment: “The introduction is too general and lacks a clear research gap analysis. What is the contribution of this paper? Where is the misunderstanding in the field? How can this add value?”
Response: The revised Introduction now clearly identifies what is unknown (the role of fatty acid ratios in determining stroke outcome) and how our study will fill that gap. This should make the purpose and value of our work explicit. - Comment: “What is the hypothesis of this paper? What are the aims and objectives of this manuscript?”
Response: The manuscript now contains a dedicated segment in the Introduction that lays out both the aims and the working hypothesis. This addition remedies the previous omission of a hypothesis. In combination with the previous point, the Introduction now ends with a strong rationale and plan for the study. - Comment: “The manuscript needs some language and grammar editing.”
Response: We have carefully edited the manuscript for grammar, syntax, and style. The narrative is now more polished and professional. Similarly, throughout the manuscript we corrected grammatical errors and improved readability. - Comment: “The manuscript has too many paragraphs. For example, in the Discussion section, some paragraphs are only 2–3 lines (lines 280-310). This breaks the flow of the manuscript.”
Response: The Discussion now has a more standard paragraph structure, each covering a distinct aspect (overall findings, sex differences, comparisons with literature, etc.), rather than a series of isolated sentences. This change improves readability — the argument in each paragraph is more developed and easier to follow. - Comment: “Table 1: age range should be provided.”
Response: The age range of the participants is now clearly indicated. This fulfills the reviewer’s request by providing additional context about the sample’s age distribution. - Comment: “Figure legends are too short and not informative.”
Response: We have made the figure legends self-explanatory. We modified Figure 2-5 to clearly state in the legend what blue and red colors mean as significance. We also added additional text and number information in the figures to add more context to the plots. - Comment: “The methods section is poorly written. The methodology lacks clarity and reproducibility.”
Response: The Methods section is now more transparent and easier to follow. It was also moved after Introduction (as per journal recommendation). By rephrasing certain sentences, we removed ambiguities and improved readability. Importantly, we added missing details that are necessary for reproducibility: how we defined the outcome (severe disability), and exactly which variables were analyzed. Collectively, these edits address the reviewer’s concerns by ensuring that another researcher could understand and replicate our methodology. We believe the Methods are now written with sufficient clarity. If there are any missing information that you think are necessary for reproducing the study please let us know. - Comment: “The inclusion and exclusion criteria are not detailed enough.”
Response: The inclusion/exclusion criteria are now clearer. Originally, some of this information was present but not clearly organized or was incomplete (for instance, age ≥18 was assumed but not stated). We have now made it explicit. A reader can see immediately who was eligible and who was left out. By stating these criteria clearly (and aligning them with what we actually did), we ensure that our methods meet the standard for reproducibility and transparency that the reviewer was looking for. - Comment: “The statistics section is not clear.”
Response: The statistical methods are now described step-by-step, which should eliminate any ambiguity. We explicitly mention which statistical tests were used for which purposes. We also defined the outcomes and list the covariates in the regression model, as noted. These additions directly address any “unclear” aspects – a reader now knows exactly how we arrived at the results. For example, if someone wondered how we handled the ordinal mRS in analysis, now it is clearer. The improvement in clarity will help others understand our analysis decisions and trust the results. - Comment: “The results are presented in a vague manner without sufficient numerical detail.”
Response: The Results section is now much more data-rich. By including the actual numbers (means, ORs, percentages, etc.), we eliminate vagueness. The reviewer will find that each result is backed with quantitative detail: for instance, instead of saying “older age was associated with lower likelihood of severe disability,” we now say “each additional year of age corresponded to an OR of 0.96 for severe disability, p<0.001,” which is precise (and we also clarified elsewhere that this finding likely reflects a complex interaction given mortality). Every important comparison or finding is now quantified, which makes the results transparent. - Comment: “The discussion is largely speculative and lacks a connection to the results. The discussion should discuss the obtained results and make connections with available literature by comparison and contrast. No comparison with previous literature or explanation of how findings align or differ.”
Response: The Discussion now sticks much closer to our data. Any claims made are supported either by our results or by references to similar findings in the literature. For instance, if previous studies showed “omega-3 benefits stroke recovery,” we explain how our findings both agree (DHA appears beneficial, consistent with those studies). By doing so, we avoid unsupported speculation and instead provide a reasoned, evidence-based discussion. We modified a big part of the discussion section in order to comply to your suggestions. - Comment: “The manuscript needs a limitations paragraph to explain the limitations.”
Response: We openly acknowledge the main limitations of our study. This limitations discussion also helps temper any strong claims, aligning the tone of our conclusions with the data. Including this section makes our manuscript more balanced and credible by showing that we understand where caution is needed in interpreting our results. - Comment: “The conclusion is too broad and doesn’t really connect to the results. It makes claims that are not fully supported by the data.”
Response: The revised conclusion is concise and directly tied to what we observed. We avoid sweeping statements. For instance, instead of saying “fatty acids might…decrease the rates of severe disability and mortality” (which was speculative), we now simply suggest fatty acid profiles could be used in risk assessment to predict disability – a claim supported by our analysis. The conclusion now also implicitly acknowledges limits by suggesting further research, which aligns with the addition of a limitations paragraph. This addresses the reviewer’s concern by ensuring we do not overstate our findings. - Comment: “Some landmark and important papers in the field are not cited.”
Response: We added several key references that were initially missing, thereby citing landmark studies relevant to our work. For example, Calder (2010, over 700 citations) https://www.mdpi.com/2072-6643/2/3/355 for the foundational knowledge on PUFA-derived mediators in inflammation. These and other references are now integrated into the Introduction and Discussion.
Round 2
Reviewer 2 Report
Comments and Suggestions for Authors
There are still many points which need to be improved, especially in Discussion.
- There are plenty of grammar and language mistakes. The manuscript needs grammar and language editing.
- Line 261: "score" should change to scale.
- Line 312: "do" should change to "did"
- Line 320: "show" should change to "showed"
- Line 345: this sentence is vague.
- Line 350: "traditional" should change to "conventional"
- Line 365: why is there emphasis on the name of the author? " Arboix" Emphasis should be on the content of the paper. Focus on authors names and studies remove attention from the results.
- Discussion doesn't have a nice flow. It should be focused to have a nice flow from one idea to another.
- Figure legends are too brief and doesn't provide a useful description.
- Table 1: all abbreviations should be introduced in table footnotes.
- Table 1: age range should be mentioned too.
- Methods: inclusion and exclusion criteria should be numbered: 1, 2, 3 to make it clear.
Author Response
We have carefully addressed each of your comments point-by-point. Below we detail our responses:
- Comment: “There are plenty of grammar and language mistakes. The manuscript needs grammar and language editing.”
Response: We thoroughly proofread and edited the language throughout the manuscript. All grammatical errors and awkward phrasings that we could identify have been corrected. For instance, we corrected verb tenses when describing past studies, fixed spelling mistakes, and replaced informal expressions with formal academic language. - Comment (Line 261): “‘score’ should change to scale.”
Response: Corrected. - Comment (Line 312): “‘do’ should change to ‘did’”
Response: Corrected. Additionally, we ensured that all references to past studies are consistently in the past tense. These tense corrections address the reviewer’s point about improper grammar. - Comment (Line 320): “‘show’ should change to ‘showed’”
Response: Corrected. - Comment (Line 345): “this sentence is vague.”
Response: Clarified. We realized the original wording was indeed unclear. We have rewritten the sentence to be more specific. This new formulation clearly conveys the cause-and-effect relationship. The revised sentences should now be easy to understand. - Comment (Line 350): “‘traditional’ should change to ‘conventional’”
Response: We have replaced the term “traditional risk factors” with “conventional risk factors” everywhere in the manuscript to maintain consistent terminology. This addresses the reviewer’s note and aligns the language with the preferred term (also used in our title and background). - Comment (Line 365): “why is there emphasis on the name of the author? ‘Arboix’… Emphasis should be on the content of the paper. Focus on authors names and studies remove attention from the results.”
Response: We have removed the emphasis on the authors’ name to ensure that whenever we discuss literature, the emphasis is on the study results and their relation to our work, rather than the authors themselves. This was applied to all paragraphs in question. - Comment: “Discussion doesn't have a nice flow. It should be focused to have a nice flow from one idea to another.”
Response: We restructured the entire section, by adding 4 subsections to group up the paragraphs according to their subject. We also rephrased several sentences/paragraphs and moved part of them to the subsection they fitted best to have the Discussion flows logically from one idea to the next. - Comment: “Figure legends are too brief and doesn't provide a useful description.”
Response: Expanded all figure legends. We have revised each figure legend to include a more informative description of the figure’s contents and key findings. These additions mean the legends are no longer just titles, but short summaries. A reader can get a sense of what each figure shows without referring back to the main text. - Comment: “Table 1: all abbreviations should be introduced in table footnotes.”
Response: Added footnotes defining abbreviations. With the new footnote, any reader consulting Table 1 can immediately understand these terms. This change directly implements the reviewer’s suggestion. - Comment: “Table 1: age range should be mentioned too.”
Response: In fact, the ages were originally presented with ranges in parentheses since the previous major review (e.g., “67.27 ± 14.14 (25–100)”), but to avoid any ambiguity, we explicitly clarified this in the table footnote as well. The footnote now notes that the values in parentheses represent the minimum and maximum ages. If it was not evident before, it is now clearly stated as per the reviewer’s request. - Comment: “Methods: inclusion and exclusion criteria should be numbered: 1, 2, 3 to make it clear.”
Response: Reformatted as numbered lists. It directly addresses the reviewer’s comment by making our inclusion/exclusion criteria clear and unambiguous.
We believe we have fully addressed all the issues raised. All the changes made were strictly in line with the suggestions, and we have not introduced any new content beyond what was recommended. The manuscript has significantly improved in clarity, flow, and readability as a result.